# Healthcare System Distrust and Non-Prescription Antibiotic Use: A Cross-Sectional Survey of Adult Antibiotic Users

**DOI:** 10.3390/antibiotics12010079

**Published:** 2023-01-01

**Authors:** Brooke A. Hawkes, Sana M. Khan, Melanie L. Bell, Jill Guernsey de Zapien, Kacey C. Ernst, Katherine D. Ellingson

**Affiliations:** 1Colorado Center for Personalized Medicine, University of Colorado Anschutz Medical Campus, Aurora, CO 80045, USA; 2Department of Epidemiology and Biostatistics, University of Arizona, Tucson, AZ 85721, USA; 3Department of Health Promotion Sciences, University of Arizona, Tucson, AZ 85721, USA

**Keywords:** antimicrobial resistance, antimicrobial stewardship, trust in healthcare system, non-prescription antibiotic use

## Abstract

Antibiotic resistance is a major public health concern driven by antibiotic overuse. Antibiotic stewardship programs are often limited to clinical settings and do little to address non-prescription antibiotic use in community settings. This study investigates the association between non-prescription antibiotic use and healthcare system distrust in the United States and Mexico. An online survey was deployed in the United States and Mexico with enhanced sampling through in-person recruiting in the border region. Non-prescription antibiotic use was defined as having bought or borrowed non-prescription oral or injectable antibiotics within the last 3 years. The survey included a previously validated 10-item scale to measure healthcare system distrust. Logistic regression was used to model the use of non-prescription antibiotics by the level of healthcare system distrust, adjusted for demographic characteristics and antibiotic knowledge. In total, 568 survey participants were included in the analysis, 48.6% of whom had used non-prescription oral or injectable antibiotics in the last 3 years. In the fully adjusted regression model, the odds of using non-prescription antibiotics were 3.2 (95% CI: 1.8, 6.1) times higher for those in the highest distrust quartile versus the lowest. These findings underscore the importance of community-based antibiotic stewardship and suggest that these programs are particularly critical for communities with high levels of healthcare system distrust.

## 1. Introduction

Antibiotic resistance is a growing public health concern that is driven primarily by overuse of antibiotics and requires urgent attention [1,2,3,4,5,6]. In the absence of concerted efforts to curb resistance, by the year 2050, resistant infections could cause 10 million deaths per year compared to the estimated 8.2 million deaths that are currently attributed to cancer each year [7]. In the United States alone, there are more than 2.8 million antimicrobial-resistant infections each year, and more than 35,000 deaths [8]. Mexico lacks a nation-wide reporting system for antimicrobial resistance; however, reports of specific pathogens suggest that a similar problem is present [9,10]. Antimicrobial stewardship programs—which promote and measure the appropriate use of antibiotics by reducing unnecessary prescribing and selection of optimal drug regimens, dose, duration of therapy, and route of admission [11]—have been implemented to address inappropriate use in clinical settings. However, since the majority of these programs are implemented through clinicians and in healthcare settings [12,13], they fail to address the issue of self-medicating through non-prescription antibiotic use. Further, these programs operate under the assumption that the public utilizes the healthcare system to acquire antibiotics, and that those who utilize the healthcare system trust their healthcare providers enough to follow recommendations.

With the exception of some over-the-counter topical antibiotics, the law in both the United States and Mexico requires antibiotic users to have a prescription from a clinician [14,15]. However, this law does not stop individuals from borrowing leftover antibiotics from family members or friends, using their own leftover supply from a previous prescription, or buying antibiotics from illegal vendors who do not require a prescription [14,16,17,18,19,20,21]. Developing a more extensive understanding of what motivates someone to self-medicate with antibiotics can help inform the development of a broader base of stewardship programs, including community-based programming. 

In both the United States and Mexico, healthcare system distrust is a potential motivator for non-prescription antibiotic use. Healthcare system distrust includes both distrust in one’s provider as well as an overarching skepticism of the medical system as a whole, including a sense of whether or not they will be able to access services when necessary. Now more than ever, public skepticism and distrust of the healthcare system, including hospitals, health insurance companies, and medical research, is persistent and continues to grow [22,23,24]. Healthcare system distrust can have a wide variety of negative consequences, including lower utilization of healthcare services, [25,26] worse management of health conditions [27,28], and lower involvement in medical research [29,30]. Since healthcare system distrust can result in lower utilization of healthcare services, there is reason to believe this could influence an individual’s decision to self-medicate with antibiotics. 

There are multiple potential confounders to consider when examining the relationship between healthcare system distrust and non-prescription antibiotic use. Previous studies have shown that young adults are the most likely to self-medicate as they become more independent and are encouraged to be independent about self-care [31,32]. Older age is also associated with healthcare system distrust [33]. Women have been prescribed antibiotics at higher rates than men [34], so men may be more likely to self-medicate. Further, previous studies have suggested that women are more likely to feel like they are not being cared for appropriately during their encounters with healthcare professionals [35]. While there are no previous studies examining the relationship between antibiotic knowledge and healthcare system distrust or non-prescription antibiotic use, there is reason to believe that poor knowledge could result in higher distrust and higher non-prescription use. Not knowing how to appropriately use antibiotics or not understanding the consequences of inappropriate use could result in using non-prescription antibiotics inappropriately and at a higher rate. Similarly, someone with low antibiotic knowledge may not trust their healthcare provider’s guidance if they do not fully comprehend the science behind the recommendations being given. While there are similarities between the United States and Mexico’s healthcare systems, there are many distinct differences that could alter both an individual’s trust in their healthcare system and their non-prescription antibiotic use. Since the healthcare systems operate differently, healthcare system distrust differs between the countries as well. Reasons for distrust and levels of distrust may vary. Nonprescription antibiotics are much more readily available in Mexico than in the United States, so this could also alter the relationship between healthcare system distrust and non-prescription antibiotic use [36]. The objective of this study is to determine if healthcare system distrust is a predictor of non-prescription antibiotic use after adjusting for potential confounders, using data from a survey of adult antibiotic users in the United States and Mexico.

## 2. Results

### 2.1. Primary Analysis

#### 2.1.1. Descriptive Statistics

For the primary analysis, 568 participants were included from the 983 original responses after the exclusion criteria were applied (Figure 1). In general, the sample was younger (78.5% < 45 years), white (70.3%), middle/high income (91%), and had at least a high school diploma (78.9%). Most were not Hispanic/Latinx or of Spanish Origin (67.4%) and had a primary residence in the United States (83.6%). Of the total sample, 10.2% lived within 100 kilometers of the United States–Mexico border. There was an even distribution of political affiliations (very left wing/liberal (9.7%), left wing/liberal (18.8%), center left/slightly liberal (9.2%), middle of the road (19.9%), center right/slightly conservative (11.1%), right wing/conservative (23.8%), very right wing/conservative (7.6%)) and only slightly more men (54.6%) than women (45.4%). The mean antibiotic knowledge score was 5.11 out of 10. Complete summary statistics are shown in Table 1.

In unadjusted comparisons, (Table 1) age, education level, income, political views, race, ethnicity, antibiotic knowledge, and healthcare system distrust had differences by non-prescription antibiotic use.

#### 2.1.2. Unadjusted Model

Before adjusting for potential confounding, we found that as healthcare system distrust increases, the odds of non-prescription antibiotic use also increase (Table 2). More specifically, the odds of someone in quartile 2 using non-prescription antibiotics is 1.65 times that of someone in quartile 1, the lowest distrust quartile (95% Confidence Interval [CI]: 1.0, 2.7). The odds of someone in quartile 3 using non-prescription antibiotics is 4.19 (95% CI: 2.6, 6.8) times that of someone in quartile 1, and the odds of someone in quartile 4 are 6.23 (95% CI: 3.6, 10.8) times that of someone in quartile 1.

#### 2.1.3. Fully Adjusted Model

After adjusting for all potential confounders, we saw a similar relationship, whereas the quartile of distrust increased, and so did the odds of non-prescription antibiotic use. Someone in quartile 2 had 1.33 (95% CI: 0.8, 2.3) times the odds of using non-prescription antibiotics, quartile 3 had 2.10 (95% CI: 1.2, 3.6) times the odds, and quartile 4 had 3.20 (95% CI: 1.8, 6.1) times the odds compared to quartile 1. 

#### 2.1.4. Partially Adjusted Model

The significant confounders that we included in our partially adjusted model included race and antibiotic knowledge scores. We also included political views in this model since it was only marginally insignificant in our fully adjusted model (95% CI: 1.0, 2.4). Our partially adjusted model yielded similar results to our unadjusted and fully adjusted models, with non-prescription antibiotic use increasing with the level of healthcare system distrust. Quartile 2 had 1.34 (95% CI: 0.8, 2.4) times the odds of using non-prescription antibiotics compared to quartile 1, quartile 3 had 2.36 (95% CI: 1.4, 4.0) times the odds, and quartile 4 had 3.51 (95% CI: 1.9, 6.5) times the odds. All three of these models are also visualized in Figure 2.

### 2.2. Sensitivity Analyses

#### 2.2.1. Sensitivity Analysis I

As shown in Table 3, the results of the first sensitivity analysis, which restricted the partially adjusted model to recent oral antibiotic users only, there were similar trends as the primary analysis. After redefining the exclusion criteria, the sample consisted of 378 responses. There was an increase in odds of using non-prescription antibiotics as the quartile of healthcare system distrust scores increased, consistent with the original findings. 

#### 2.2.2. Sensitivity Analysis II + III

Table 4 shows the results of the second and third sensitivity analyses, which reveal that the magnitude of the effect varies depending on whether “non-prescription antibiotic use” is defined as buying non-prescription antibiotics or borrowing non-prescription antibiotics from a family member or friend. When it is defined as buying non-prescription antibiotics, the odds ratios are smaller. Using this definition, someone in quartile 2 has 1.23 (95% CI: 0.7, 2.2) times the odds of using non-prescription antibiotics compared to quartile 1, someone in quartile 3 has 1.65 (95% CI: 0.9, 2.9) times the odds, and someone in quartile 4 has 1.97 (95% CI: 1.1, 3.6) times the odds. However, when “non-prescription antibiotic use” is defined as borrowing non-prescription antibiotics from a family member or friend, someone in quartile 2 has 2.56 (95% CI: 1.2, 5.3) times the odds of using non-prescription antibiotics compared to quartile 1, someone in quartile 3 has 3.33 (95% CI: 1.7, 6.7) times the odds, and someone in quartile 4 has 5.95 (95% CI: 2.8, 12.5) times the odds. 

## 3. Discussion

This study demonstrated that individuals with high levels of healthcare system distrust have higher odds of using non-prescription antibiotics compared to those with lower levels of healthcare system distrust. Notably, almost half of the study sample (48.6%) had used non-prescription antibiotics, revealing how prevalent the practice of antibiotic use outside the sanction of the medical system is today. Race and antibiotic knowledge remained significant in the adjusted models, and political views was marginally significant. These findings indicate that the relationship between healthcare system distrust and non-prescription antibiotic use is complex, and these significant covariates could be potential predictors to explore further in future research. After redefining “non-prescription antibiotic use” to either buying non-prescription antibiotics or borrowing them from a family member or friend, the effect of healthcare system distrust on non-prescription antibiotic use was greater for those who borrowed from a family member or friend. Further exploring the nuances to how individuals are choosing to access antibiotics is an important next step in future research. 

This study was one of the first of its kind in identifying antibiotic use and practices on this scale. Many prior non-prescription antibiotic use studies have focused on specific populations [18,37,38,39,40,41,42]; however, this study was much more generalizable. It was one of the first to examine the motivations behind non-prescription antibiotic use which is information that can be leveraged to further the reach of antimicrobial stewardship programs. The cross-sectional design allowed this study to be conducted relatively quickly and inexpensively. The binational nature of this study was also a strength. Since antibiotic resistance is a global health problem, it is important that the research surrounding it is not siloed and that we continue the global collaboration. Seeing how practices in Mexico are impacting the United States, and vice versa, allows us to treat this issue more broadly.

The biggest limitation of this study was that the study sample was not well distributed between Mexico and United States participants, which is a deviation from our sampling goals. The sample was mostly from the United States, and therefore the study sample was not as representative of Mexico as it was of the United States. According to responses in our survey, participants from Mexico obtained antibiotics primarily from community pharmacies, family members or friends, health stores, and medical clinics. Prior literature suggests that prescriptions for antibiotics in Mexico are very easy to obtain, as physician consultation offices have started to emerge within pharmacies [43]. The United States has recently focused more efforts and resources on lowering the prescription of antibiotics from clinicians [12,13], so the culture of obtaining antibiotics between the two countries is different. Our United States sample mostly obtained antibiotics from family members or friends, internet pharmacies or online, or by using their own leftover supply from a previous sickness. Because of the differences in obtaining antibiotics by country, it is possible that the underrepresentation from Mexico participants influenced our results to more closely reflect practices occurring in the United States. The survey was solely online, which could have excluded individuals without access to the internet. Along the border, flyers were distributed at health clinics and COVID-19 testing sites which could have resulted in selection bias, since everyone along the border who took the survey had learned about the survey from a flyer posted at a healthcare site. Recruitment may have been more effective in individuals with a high trust in the healthcare system in the border region since those with higher levels of distrust would be less likely to be utilizing their healthcare system [25]. Finally, this cross-sectional survey was distributed over 10 months during a pandemic, and it is possible that individual responses would have differed depending on when they took the survey during that time period. 

Prior literature suggests that non-prescription antibiotic use is not a phenomena limited to this study alone [14,17,18,19,20,21,21,37,38]. Further research is needed to fully understand why individuals choose to access antibiotics outside of the medical system. While this study revealed that healthcare system distrust is a potential motivator, this is a complex and nuanced issue that cannot be fully addressed with a single cross-sectional study. Diversifying the populations being studied will help us develop a more well-rounded understanding of how non-prescription antibiotic use is affecting antibiotic resistance globally and how motivations differ or are similar across different cultures and regions of the world. The healthcare system distrust scale was developed for the United States healthcare system, so using a distrust scale that is more appropriate for the Mexican healthcare system is important for fully understanding how distrust is motivating non-prescription antibiotic use outside of the United States.

It is important that antimicrobial stewardship programs are meeting people where they are, regardless of beliefs or healthcare practices. The relationship between healthcare system distrust and nonprescription antibiotic use reveals how important it is that appropriate antibiotic use messaging and AMR education are not solely limited to clinical settings. Community-based programming is an important step in bridging the gap between antimicrobial stewardship programs and non-prescription antibiotic use. Antimicrobial stewardship programs need to acknowledge that people make decisions outside of the healthcare system, and that trust is a big driver of this practice. We must avoid relying on language such as “talk to your provider” when delivering education on antibiotic use and antibiotic resistance. Further research should be done to assess how community-based programming might be leveraged to improve antimicrobial stewardship programs most effectively.

## 4. Materials and Methods

### 4.1. Study Design and Recruitment

Researchers from the University of Arizona and University of Sonora-Hermosillo designed an online survey to understand antibiotic seeking in the United States and Mexico during the COVID-19 pandemic. Recruitment throughout the United States and Mexico was done through Amazon Mechanical Turk, a website that allows individuals to respond to surveys for compensation [44]. Surveys were completed through Amazon from August 2020 through June 2021 and participants received five dollars. Oversampling at the United States–Mexico border was achieved by directly recruiting participants with flyers distributed through academic-community partnerships in the border regions. Community partners posted flyers in both English and Spanish at health clinics and COVID-19 test sites in the border region, defined as 100 kilometers from the US–Mexico border. Flyers had a Quick Response (QR) code linking participants to an online survey, which they could take in English or Spanish. Direct recruitment in the border region occurred from December 2020 to August 2021 and participants were emailed a five-dollar Amazon gift card. A stratified sampling technique was used with the goal of surveying 1000 individuals evenly across the following strata: country of residence (with at least 250 within 100 kilometers of the border), age group, and gender. The survey platform allowed us to distribute surveys within these strata. If target strata could not be filled by the last month of the study, any person was eligible to take the survey. The survey platform allowed distribution within these strata. If target strata could not be filled by the last month of the study, any person was eligible to take the survey. The English version of the full survey distributed is listed in Appendix A. All participants provided informed consent prior to taking the survey. The study protocol was reviewed and approved by the Institutional Review Board of the University of Arizona (Protocol number: 2007883915).

### 4.2. Study Sample

Surveys were deployed to individuals over the age of 18 living in the United States or Mexico. The analysis was restricted to antibiotic users in the past three years, excluding those who used topical antibiotics only as they can be legally obtained over the counter. To account for any insincere responses, anyone who had a completion time of fewer than seven minutes, decided a priori, was excluded. Participants were also eliminated if they were missing data for questions concerning non-prescription antibiotic use or demographics. To prevent answers from internet bots, anyone who incorrectly answered the question, “Please select C to make sure you are human”, was also removed. See Figure 1 for details on the study’s inclusion criteria. 

### 4.3. Variables of Interest

Non-prescription antibiotic use was defined by using two true or false questions from the survey: “I have bought non-prescription oral or injectable antibiotics within the past 3 years” and “I have asked my friends or family for leftover oral or injectable antibiotics within the past 3 years.” If a participant responds true to either of the questions, they will be considered to have sought out non-prescription antibiotics. If they answer false for both questions, they will not fall under this classification. 

The primary exposure of interest, healthcare system distrust, was measured using a 10-question scale. This scale was validated in previous literature using qualitative methods such as focus groups, pilot testing, and a cross-sectional telephone survey [45]. Each question could be answered via a scale of strongly disagree, disagree, not sure, agree, and strongly agree. Individual questions could be scored 1 to 5, with 5 being equivalent to the highest level of distrust. If a question in this scale was left blank, it was given the same number of points as answering, “not sure.” Total scores can range from 10 to 50, with higher scores indicating more distrust. Details on the questions in this scale and possible responses are provided as Appendix A. For analysis, healthcare system distrust scores were divided into quartiles, which is consistent with previously published studies in the scientific literature [33]. 

Potential covariates were selected for the adjusted model a priori based on past knowledge and a review of the literature [34,46,47,48]. Categorical variables included gender, age, education level, ethnicity, race, political affiliation, income, and proximity to the United States–Mexico border. Variable categories were collapsed if strata were thin (less than 10) including age, education level, income, and race. The continuous variable, antibiotic knowledge (scale 0–10), was a scale created by compiling questions related specifically to antibiotic knowledge on the survey. This scale was composed of 10 questions with higher scores indicating a higher level of antibiotic knowledge. The list of questions, possible responses, and how each response was scored are provided as Appendix A.

### 4.4. Statistical Analysis

For the sample of interest, descriptive statistics were calculated by stratifying on non-prescription antibiotic use history and compared using chi-square or t-tests. Unadjusted and adjusted logistic regression models were ran to estimate the odds (with 95% confidence intervals [CIs]) of non-prescription antibiotic use in each distrust quartile, using the lowest quartile as a reference group. The fully adjusted model included the covariates chosen a priori, described above. A partially adjusted model was also included, only including covariates that were statistically significant in the fully adjusted model. Statistical significance was defined as having a confidence interval that did not include 1.0.

Three sensitivity analyses were run. First, to ensure results were consistent with current antibiotic regulations, the inclusion criteria were redefined to only individuals who had used antibiotics within the year they had completed their survey. The second and third sensitivity analyses separated the outcome variable into those who borrowed non-prescription antibiotics from family or friends from those who purchased them. First, non-prescription antibiotic users were classified as anyone who answered “True” to the question “I have bought non-prescription oral or injectable antibiotics within the past 3 years” and anyone who answered “False” was classified as a prescription antibiotic user. In the next analysis, anyone who answered “True” to the question “I have asked my friends or family for leftover oral or injectable antibiotics within the past 3 years,” was classified as a non-prescription antibiotic user. All statistical analyses performed in this study used SAS OnDemand for Academics [49].

## 5. Conclusions

This study reveals that healthcare system distrust is a potential motivator for seeking out antibiotics outside of the medical system, especially when individuals are borrowing non-prescription antibiotics from a family member or friend. In all models, including those adjusted for demographic factors and knowledge about antibiotics, healthcare system distrust was positively correlated with non-prescription antibiotic use. With non-prescription antibiotic use being so common, it is crucial that antimicrobial stewardship programs expand their reach to include community-based programming outside of clinical settings. Further research should be conducted to further understand the complexities of healthcare system distrust and non-prescription antibiotic use as well as the development and implementation of community-based antimicrobial stewardship programs.

## Figures and Tables

**Figure 1 antibiotics-12-00079-f001:**
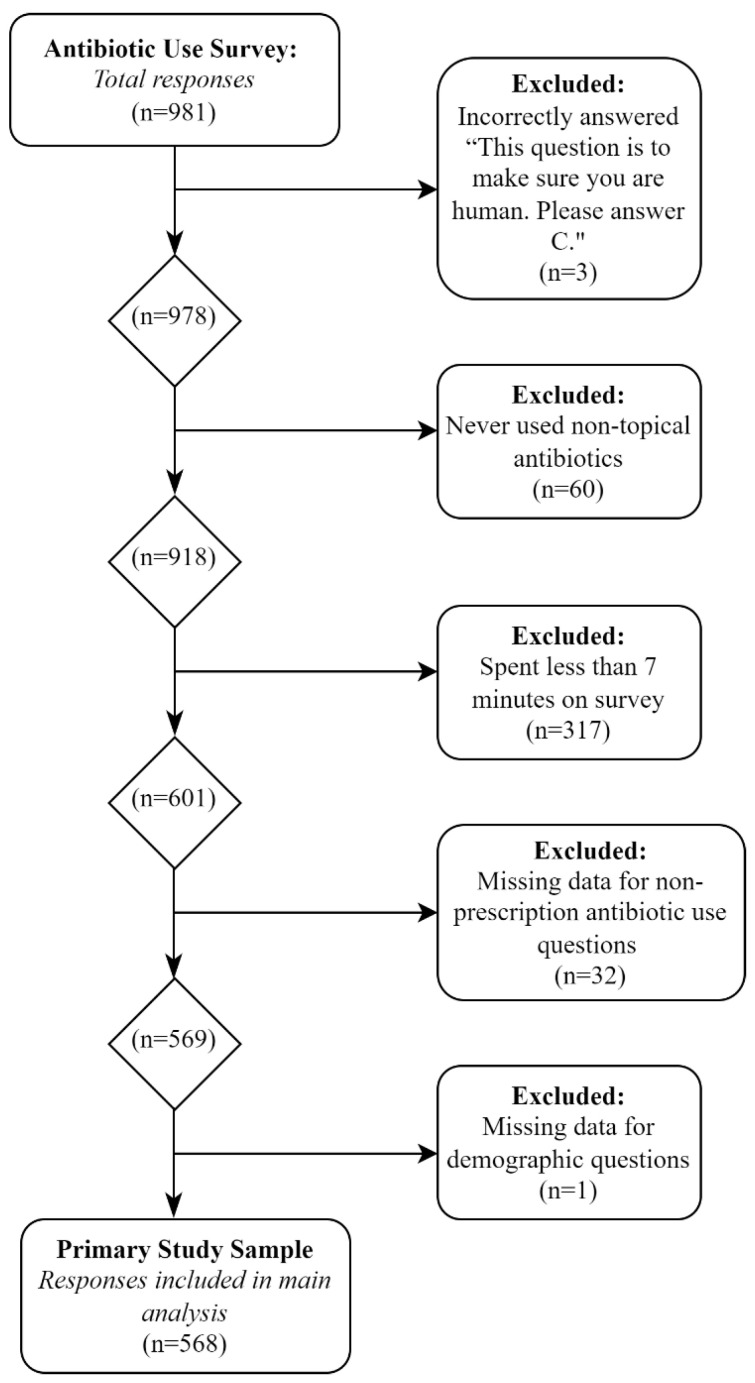
Inclusion of participants for statistical analysis.

**Figure 2 antibiotics-12-00079-f002:**
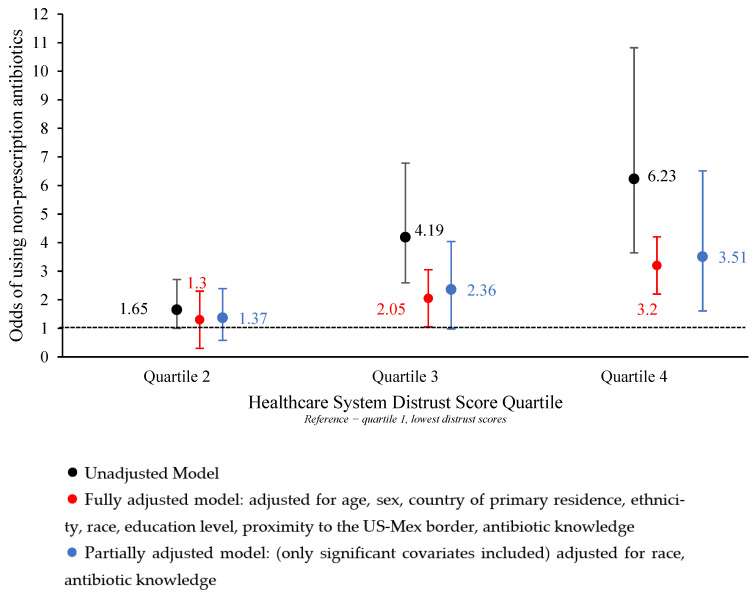
Odds of using non-prescription antibiotics by healthcare system distrust score quartile (n = 568).

**Table 1 antibiotics-12-00079-t001:** Results of logistic regression analysis for non-prescription antibiotic use with healthcare system distrust as a predictor (n = 568).

Characteristic	Total (n = 568)	Non-Prescription Antibiotic Use (n = 276)	Prescription Antibiotic Use (n = 292)	*p*-Value
	n (% ^a^)	n (% ^b^)	n (% ^b^)	χ^2^-test
**Gender**				0.57
Male	310 (54.6)	154 (49.7)	156 (50.3)	
Female	258 (45.4)	122 (47.3)	136 (52.7)	
**Age**				<0.01
18–24	28 (4.9)	15 (53.8)	13 (46.4)	
25–34	229 (40.3)	132 (57.6)	97 (42.4)	
35–44	189 (33.3)	87 (46.0)	102 (54.0)	
45–54	66 (11.6)	24 (36.4)	42 (63.6)	
55+	56 (9.9)	18 (32.1)	38 (67.9)	
**Education level**				<0.01
High school diploma or less	120 (21.1)	44 (36.6)	76 (63.3)	
High school diploma or more	448 (78.9)	232 (51.8)	216 (48.2)	0.02
**Income ^c^**				
Low	51 (9.0)	26 (51.0)	25 (49.0)	
Middle	301 (53.0)	161 (53.5)	140 (46.5)	
High	216 (38.0)	89 (41.2)	127 (58.8)	
**Political views**				<0.01
Very left wing/liberal	55 (9.7)	15 (27.3)	40 (72.7)	
Left wing/liberal	107 (18.8)	51 (47.7)	56 (52.3)	
Center left/slightly liberal	52 (9.2)	23 (44.2)	29 (55.8)	
Middle of the road	113 (19.9)	40 (35.4)	73 (64.6)	
Center right/slightly conservative	63 (11.1)	29 (46.0)	34 (54.0)	
Right wing/conservative	135 (23.8)	89 (65.9)	46 (34.1)	
Very right wing/conservative	43 (7.6)	29 (67.4)	14 (32.6)	
**Ethnicity**				0.15
Hispanic/Latinx	185 (32.6)	98 (53.0)	87 (47.0)	
Non-Hispanic/Latinx	383 (67.4)	178 (46.5)	205 (53.5)	
**Race**				<0.01
White	399 (70.3)	180 (45.1)	219 (54.9)	
American Indian or Alaskan Native	11 (1.9)	6 (54.6)	5 (45.5)	
Asian	24 (4.2)	9 (37.5)	15 (62.5)	
Black or African American	79 (13.9)	58 (73.4)	21 (26.6)	
Native Hawaiian or Pacific Islander	6 (1.1)	6 (100.0)	0 (0.0)	
Mixed race	20 (3.5)	6 (30.0)	14 (70.0)	
Other	29 (5.1)	11 (37.9)	18 (62.1)	
**Country of primary residence**				0.62
United States	475 (83.6)	233 (49.0)	242 (51.0)	
Mexico	93 (16.4)	43 (46.2)	50 (53.8)	
**Proximity to the US-Mex border**				0.82
Zip code ≤ 100 km	58 (10.2)	29 (50.0)	29 (50.0)	
Zip code >100 km	510 (89.8)	247 (48.4)	263 (51.6)	
**Healthcare system distrust**				<0.01
Quartile 1 (lowest scores)	143 (25.2)	39 (27.3)	104 (72.7)	
Quartile 2	144 (25.4)	55 (38.2)	89 (61.8)	
Quartile 3	167 (29.4)	102 (61.1)	65 (38.9)	
Quartile 4 (highest scores)	114 (20.1)	80 (70.2)	34 (29.8)	
	mean (±SD)	mean (±SD)	mean (±SD)	*t*-test
**Healthcare system distrust ^d^**	28.3 (±6.7)	30.6 (±6.1)	26.1 (±6.8)	<0.01
**Antibiotic knowledge ^e^**	5.11 (±3.0)	3.6 (±2.5)	6.5 (±2.8)	<0.01

^a^ Column %; ^b^ Row %; ^c^ Low income: <USD 20,000 (US)/<USD 10,000 (pesos) (MX); Middle: USD 20,000–60,000 (US)/USD 10,000–30,000 (pesos) (MX); High: USD 60,000+ (US)/USD 30,000 (pesos) (MX); ^d^ Scores possible: 10–50, higher scores = more healthcare system distrust; ^e^ Scores possible: 0–10, higher scores = more antibiotic knowledge; SD = standard deviation.

**Table 2 antibiotics-12-00079-t002:** Results of logistic regression analysis for non-prescription antibiotic use with healthcare system distrust as a predictor (n = 568).

	Unadjusted OR (95% CI)	Fully Adjusted ^a^ OR (95% CI)	Partially Adjusted ^b^ OR (95% CI)
**Healthcare system distrust score ^c^**			
Quartile 1 (lowest)	1.00 (ref)	1.00 (ref)	1.00 (ref)
Quartile 2	1.65 (1.0, 2.7)	1.33 (0.8, 2.3)	1.34 (0.8, 2.4)
Quartile 3	4.19 (2.6, 6.8)	2.10 (1.2, 386)	2.36 (1.4, 4.0)
Quartile 4 (highest)	6.23 (3.6, 10.8)	3.20 (1.8, 6.1)	3.51 (1.9, 6.5)
**Gender**			
Male	-	1.00 (ref)	-
Female	-	1.09 (0.7, 1.6)	-
**Age**			
18–34	-	1.00 (ref)	-
35+	-	0.72 (0.5, 1.1)	-
**Education level**			
High school diploma or more	-	1.00 (ref)	-
High school diploma or less	-	0.83 (0.5, 1.4)	-
**Income ^d^**			
High	-	1.00 (ref)	-
Low/middle	-	1.45 (0.9, 2.2)	-
**Political views**			
Liberal/middle of the road	-	1.00 (ref)	1.00 (ref)
Conservative	-	1.56 (1.0, 2.4)	1.39 (0.9, 2.1)
**Ethnicity**			
Not Hispanic/Latinx or of Spanish origin	-	1.00 (ref)	-
Hispanic/Latinx or of Spanish origin	-	1.16 (0.7, 2.0)	-
**Race**			
Non-black/African American	-	1.00 (ref)	1.00 (ref)
Black/African American	-	2.34 (1.3, 4.3)	2.48 (1.4, 4.5)
**Country of primary residence**			
United States	-	1.00 (ref)	-
Mexico	-	1.40 (0.7, 2.9)	-
**Proximity to the US-Mex border**			
Zip code > 100 km	-	1.00 (ref)	-
Zip code ≤ 100 km	-	1.27 (0.6, 2.6)	-
**Antibiotic knowledge ^e^**	-	0.75 (0.7, 0.8)	0.75 (0.7, 0.8)

OR = odds ratio, CI = confidence interval; ^a^ Adjusted for: gender, age, education level, income, political views, ethnicity, race, country of primary residence, proximity to US-MEX border, antibiotic knowledge level; ^b^ Adjusted for only significant covariates: race, political views, antibiotic knowledge; ^c^ Scores possible: 10–50, higher scores = more healthcare system distrust; ^d^ Low/middle income: <USD 60,000 (US)/<USD 30,000 (pesos) (MX); High: USD 60,000+ (US)/USD 30,000 (pesos) (MX); ^e^ Scores possible: 0–10, higher scores = more antibiotic knowledge.

**Table 3 antibiotics-12-00079-t003:** Results of primary logistic regression analysis compared to sensitivity analysis, restricted to recent antibiotic users only.

	Primary Analysis (n = 568) Adjusted OR ^a^ (95% CI)	Sensitivity Analysis (n = 387) Adjusted OR ^a^ (95% CI)
**Healthcare system distrust score ^b^**		
Quartile 1 (lowest)	1.00 (ref)	1.00 (ref)
Quartile 2	1.34 (0.8, 2.4)	1.91 (1.0, 3.8)
Quartile 3	2.36 (1.4, 4.0)	2.53 (1.3, 4.9)
Quartile 4 (highest)	3.51 (1.9, 2.1)	3.29 (1.5, 7.0)
**Political views**		
Liberal/middle of the road	1.00 (ref)	1.00 (ref)
Conservative	1.39 (0.9, 2.1)	1.36 (0.8, 2.3)
**Race**		
Non-black/African American	1.00 (ref)	1.00 (ref)
Black/African American	2.48 (1.4, 4.5)	1.66 (0.9, 3.2)
**Antibiotic knowledge ^c^**	0.75 (0.7, 0.8)	0.75 (0.7, 0.8)

OR = odds ratio, CI = confidence interval; Recent antibiotic users = individuals who answered: “In the last year”, “In the last 6 months”, “Within the past 3 months”, or “Within the past month” to the question “When was the last time you took oral antibiotics outside of a hospital setting?”; ^a^ Adjusted for only significant covariates: race, political views, antibiotic knowledge; ^b^ Scores possible: 10–50, higher scores = more healthcare system distrust; ^c^ Scores possible: 0–10, higher scores = more antibiotic knowledge.

**Table 4 antibiotics-12-00079-t004:** Results of sensitivity analyses: relationship between healthcare system distrust and non- prescription antibiotic use, non-prescription antibiotic use redefined (n = 568).

	Bought Non-Prescription Antibiotics ^a^ (n = 568) Adjusted OR ^c^ (95% CI)	Borrowed Non-Prescription Antibiotics ^b^ (n = 568) Adjusted OR ^c^ (95% CI)
**Healthcare system distrust score ^d^**		
Quartile 1 (lowest)	1.00 (ref)	1.00 (ref)
Quartile 2	1.23 (0.7, 2.2)	2.56 (1.2, 5.3)
Quartile 3	1.65 (0.9, 2.9)	3.33 (1.7, 6.7)
Quartile 4 (highest)	1.97 (1.1, 3.6)	5.95 (2.8, 12.5)
**Political views**		
Liberal/middle of the road	1.00 (ref)	1.00 (ref)
Conservative	0.92 (0.6, 1.4)	2.20 (1.4, 3.4)
**Race**		
Non-black/African American	1.00 (ref)	1.00 (ref)
Black/African American	2.79 (1.7, 4.7)	1.72 (1.0, 3.0)
**Antibiotic knowledge** ** ^e^ **	0.78 (0.7, 0.8)	0.75 (0.7, 0.8)

OR = odds ratio, CI = confidence interval; ^a^ Non-prescription antibiotic use redefined: “True” for the question “I have bought non-prescription oral or injectable antibiotics within the past 3 years”; ^b^ Non-prescription antibiotic use redefined: “True” for the question “I have asked my friends or family for leftover oral or injectable antibiotics within the past 3 years”; ^c^ Adjusted for only significant covariates: race, political views, antibiotic knowledge; ^d^ Scores possible: 10–50, higher scores = more healthcare system distrust; ^e^ Scores possible: 0–10, higher scores = more antibiotic knowledge.

## Data Availability

The data available in this study are available on request from the corresponding author. The data are not publicly available due to being the property of the University of Arizona and the Arizona Area Education Center.

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
