# Peer review of "Healthcare System Distrust and Non-Prescription Antibiotic Use: A Cross-Sectional Survey of Adult Antibiotic Users"

_antibiotics, 2023, doi:10.3390/antibiotics12010079_

Round 1

Reviewer 1 Report

1. Please bring some more conclusive messages from the data collected through the questionnaire.

2. The conclusion part of the manuscript can be improved.

3. The ethical approval details can be included since the law violations were reported in the study.

Reviewer 2 Report

Line 11 and line 30, Antimicrobial resistance is a public health threat?? use another word in place of threat.

What is the relation of time and response in your study?

Access of antibiotics by the patients? categorization is missing.

Assessment of resistance based on online survey. How?

What is confidence of interval based on patients' responses?

Any consent with the attached questionnaire. 

Antibiotics access specially in USA is not easy?? explain more how they get these antibiotics?

What is the relation of Healthcare System Distrust with your study?

Reviewer 3 Report

I have enjoyed reading article entitled “Healthcare System Distrust and Non-Prescription Antibiotic Use: a cross-sectional sur- 2 vey of adult antibiotic users” by Hawkes et al.  There are few comments and suggestions that need to be considered before further consideration of this manuscript.

Introductions

Authors should provide information in detail about the magnitude of the ABR problem in the UK and Mexico.

Regulations regarding non-prescription antibiotic use should be elaborated.

Discuss the rationale of your study topic.

Methods

No information about the sample size has been provided for both the UK and Mexico.

What was the sampling technique?

How have you determined the validity of your questionnaire?

Have you measured reliability of the tool?

How Healthcare System Distrust Scale was validated? No information has been presented.

How a respondent will answer this question if he has only used oral antibiotic. “Where did you obtain the topical antibiotic and for what purpose were you taking it?” this issue is valid for rest of the questions related to injectable antibiotic use.

Discussion

This section should be backed by existing literature.

Conclusion

It should be precise and accurate. There is no need to provide introductory information regarding the AMR.

Round 2

Reviewer 1 Report

Thanks to the authors for considering the changes requested. All queries were satisfied.

Author Response

Thank you for your rereview.

Reviewer 2 Report

After revision, paper may be accepted. 

Author Response

Thank you for your rereview.

Reviewer 3 Report

The authors have addressed most of my comments. My only concern is the sample size. How authors have calculated the sample size for this study. This is important as to determine the generalizability of the findings of your study. 

Author Response

Thank you for your review of this manuscript. Below is our response to your concerns about the sample methods, which we agree needed further explanation in the methods and discussion.

Reviewer Comment: The authors have addressed most of my comments. My only concern is the sample size. How authors have calculated the sample size for this study. This is important as to determine the generalizability of the findings of your study. 

Author Response: This is an important point. For context, this was a pilot study financed by an international research development grant. Because the survey was developed with the general objective of understanding antibiotic use in the US and Mexico, we did not power based on a specific hypothesis. We aimed to sample 1000 participants with equal numbers of participants from the US and Mexico, with even age and sex distributions. We determined compensation to be $5 per survey based on participants the anticipated length of time needed to take the survey coupled with minimum wage. Amazon MTurk allows for survey distribution to be restricted based on specified demographic criteria. Due to the demographics of Amazon MTurk users, some strata were difficult to fill (e.g. above 50 years of age and living within 100 miles of the border in Mexico), which is why we conducted oversampling of individuals living within 100 miles of the border through in-person recruiting using flyers posted in clinics. These considerations indeed make our findings not necessarily generalizable.

To address these issues, we have added the following to the Materials and Methods section under the subsection “Study Sample” (new texted highlighted below and in the upload revision): A stratified sampling technique was used with the goal of surveying 1000 individuals evenly across the following strata: country of residence (with at least 200 within 100 miles of the border), age group, and gender. The survey platform allowed us to distribute surveys within these strata. If target strata could not be filled by the last month of the study, any person was eligible to take the survey.

Further, we have expounded on this limitation in the Discussion (paragraph 3 on limitations) with the addition of the following language: The biggest limitation of this study was that the study sample was not well distributed between Mexico and United States participants, which is a deviation from our sampling goals. <the text then goes on to discuss how the context differs and the impact on generalizability>